# Glucocorticoid Regulates the Synthesis of Porcine Muscle Protein through m^6^A Modified Amino Acid Transporter *SLC7A7*

**DOI:** 10.3390/ijms23020661

**Published:** 2022-01-10

**Authors:** Wei-Jing Xu, Kai Guo, Jia-Li Shi, Chang-Tong Guo, Jia-Le Xu, Rong Zheng, Si-Wen Jiang, Jin Chai

**Affiliations:** 1Agricultural Ministry Key Laboratory of Swine Breeding and Genetics & Key Laboratory of Agricultural Animal Genetics, Breeding, and Reproduction of Ministry of Education, Huazhong Agricultural University, Wuhan 430070, China; q2385685069@163.com (W.-J.X.); 15038027365@163.com (K.G.); realscary@foxmail.com (J.-L.S.); Gct792808035@163.com (C.-T.G.); xujialenice@163.com (J.-L.X.); zhengrong@mail.hzau.edu.cn (R.Z.); jiangsiwen@mail.hzau.edu.cn (S.-W.J.); 2The Cooperative Innovation Center for Sustainable Pig Production, Wuhan 430070, China

**Keywords:** pig, glucocorticoid, m^6^A, protein deposition, *SLC7A7*

## Abstract

The occurrence of stress is unavoidable in the process of livestock production, and prolonged stress will cause the decrease of livestock productivity. The stress response is mainly regulated by the hypothalamic-pituitary-adrenal axis (HPA axis), which produces a large amount of stress hormones, namely glucocorticoids (GCs), and generates a severe impact on the energy metabolism of the animal body. It is reported that m^6^A modification plays an important role in the regulation of stress response and also participates in the process of muscle growth and development. In this study, we explored the effect of GCs on the protein synthesis procession of porcine skeletal muscle cells (PSCs). We prove that dexamethasone affects the expression of *SLC7A7*, a main amino acid transporter for protein synthesis by affecting the level of m^6^A modification in PSCs. In addition, we find that *SLC7A7* affects the level of PSC protein synthesis by regulating the conduction of the mTOR signaling pathway, which indicates that the reduction of *SLC7A7* expression may alleviate the level of protein synthesis under stress conditions.

## 1. Introduction

Stress causes significant losses to the pork industry by affecting animal physiology and reducing their production performance. A previous paper has shown that compared with a control group, pigs showed a markedly reduced final body weight, average daily gain, average daily feed intake and the ratios of the feed intake to weight gain under chronic oxidative stress condition [1]. Under a stress condition, the body secretes a cascade of glucocorticoids (GCs) through the HPA axis, which is involved in many metabolic processes, such as immune system, reproduction, behavior, and cognitive function [2]. As a primary target organ, skeletal muscle is vulnerable to the effects of GCs, resulting in reduced synthesis and increased degradation of muscle proteins, leading to skeletal muscle atrophy and impaired meat quality [3,4,5].

The synthesis of muscle proteins needs free amino acids as substrates [6], and amino acid transporters are required in this process [7]. y+L+ amino acid transporter-1 (y+LAT1) is encoded by the SLC7A7 gene, and the substrates of this system are arginine, lysine, leucine, glutamine and methionine [8]. The expression level of SLC7A7 is high in intestine and kidney but low in skeletal muscle, and the mutation of SLC7A7 gene usually induces many complications, such as persistent hypotonia, birth retardation, osteoporosis, coma and mental development disorder [9,10]. It has been demonstrated that *SLC7A7^−/−^* mice exhibit intrauterine growth restriction and persistent growth retardation that generates fetal growth retardation through downregulation of IGF1, thus resulting in mouse death [11]. Similarly, *SLC7A7^−/−^* mice had significantly lower weights of liver and skeletal muscle compared with control littermates [12].

The regulation of m^6^A is particularly critical under stress conditions such as extreme temperature, hypoxia nutrient deprivation and toxin exposure due to short- or long-term effects brought about by stress [13]. m^6^A plays an important role in body when subjected to various stresses, and researches have shown that m^6^A levels in the liver and abdominal fat of offspring piglets were higher than those in heat stress conditions, and heat stress also increased the expression of genes related to adipogenesis [14]. It is also possible that METTL3 targets HSP90 and HSP70 to increase their mRNA levels in response to heat stress [15]. m^6^A is mainly involved with three proteases: methyltransferases, demethylase and readers. m^6^A methyltransferases mainly include METTL3, METTL14 and WTAP. Among them, METTL3 serves as a methyl donor, METTL14 forms a stable heterodimer with METTL3 and WTAP is responsible for recruiting METTL3 and METTL14 to form a stable complex [16,17]. Besides, m^6^A can be removed by ALKBH5 and FTO demethylases. mRNAs with m^6^A modification can be recognized by specific recognition proteins, such as YTHDF1, YTHDF2, YTHDC1, etc. Recognition proteins bind to methylated mRNA transcripts and regulate their stability, degradation and translation of mRNA [18].

In this study, we further analyzed the effects of stress on m^6^A modification and the expression of the amino acid transporter *SLC7A7* and studied the molecular mechanisms regulated by *SLC7A7* in protein synthesis in porcine skeletal muscle.

## 2. Results

### 2.1. Effects of Glucocorticoid (GC) and RU486 on Protein Synthesis in Porcine Skeletal Muscle Cells

We found that the mRNA expression levels of genes such as *Atrogin*-1, *MuRF*, *FoxO1* and *MSTN*, which are related to protein breakdown and synthesis, showed upregulation when treated with 0.1μΜ DEX in PSCs for 48 h. Among them, the protein expression levels of Atrogin-1, FoxO1 and MSTN were significantly upregulated according to the RT-PCR and Western blot results (Figure 1A–C). To further explore the effect of GCs on protein synthesis, we used a sunset non-radioactive (SUnSET) method to examine the protein synthesis of cells after treatment of DEX and its antagonist RU486, which shows that the protein synthesis rate decreased after adding DEX but then increased with the addition of RU486 (Figure 1D,F). The synthesis of muscle proteins requires activation of the mTOR signaling pathway. We then extracted proteins after treatment with DEX for 48 h in PSCs and found that the phosphorylation levels of mTOR as well as two key functional downstream proteins, S6K1 and 4EBP1, were significantly downregulated after treated with DEX. These results showed an increase in the expression levels of protein breakdown genes and a decrease in the expression levels of protein synthesis genes under stress condition in PSCs (Figure 1E,G).

### 2.2. Effects of GCs and RU486 on Intracellular m^6^A Levels in Porcine Skeletal Muscle Cells

To confirm whether GCs and their antagonist RU486 are involved in the regulation of intracellular m^6^A levels affecting the expression of intracellular genes after treatment with DEX in PSCs, we used dot blot hybridization and high-performance liquid chromatography-mass spectrometry methods and found a reduction of overall m^6^A modification level in cells after treatment with DEX for 48 h while then restored to original level after adding with RU486 (Figure 2A), and mass spectrometry assay showed the same result (Figure 2B). To gain further insight into the involvement of DEX in the regulation of m^6^A modification, we treated the cells with DEX for 48 h and then examined the expression levels of m^6^A modification enzymes, and found that the mRNA and protein expression levels of METTL3 and METTL14 were significantly downregulated while the expression of the recognition protein YTHDF2 was significantly upregulated (Figure 2C–E). The results implied that m^6^A levels were downregulated in porcine skeletal muscle cells under stress conditions likely through METTL3, METTL14 as well as the YTHDF2 protein.

### 2.3. Effect of GCs on the Amino Acid Transporter SLC7A7

We extracted RNA after DEX treatment for 48h in porcine skeletal muscle cells and used QPCR to determine the mRNA expression levels of relevant amino acid transporters, which revealed a highly significant downregulation of the mRNA expression levels of the amino acid transporters *SLC1A3*, *SLC7A7*, and *SLC7A10* (Appendix A). The expression level of *SLC7A7* mRNA was found to be greatly decreased using cortisol-fed samples from piglets (Figure 3A). *SLC7A7* mRNA expression levels as well as the protein expression levels were found to be significantly downregulated after treated with DEX for 24 h and 48 h in PSCs (Figure 3B–D). To further explore whether *SLC7A7* regulated protein synthesis by means of m^6^A modification, we predicted m^6^A single base sites of *SLC7A7* gene using the online database SRAMP, and the results showed a high confidence of three m^6^A modification sites in the *SLC7A7* gene cDNA sequence (Appendix A). Then we examined the m^6^A modification levels of *SLC7A7* after treating DEX in PSCs and found a significant decrease of *SLC7A7* compared with the control group, demonstrating that its expression was subjected to m^6^A post transcriptional modification mediated by DEX (Figure 3E). The stability of *SLC7A7* mRNA was later examined by treating PSCs with the pan-transcriptional inhibitor ActD. After treating cells with DEX for 48 h and then adding ActD for 0, 2, and 4 h respectively, we found that the expression of *SLC7A7* was significantly downregulated using QPCR, demonstrating that DEX treatment to the cells for 48 h accelerated the decay of *SLC7A7* mRNA (*p* < 0.05) (Figure 3F). We used dot blot hybridization to detect the m^6^A methyl modification level after treating PSCs with 3-deazaadenosine (DAA) for 48 h, an inhibitor of m^6^A methylation (Figure 3G). QPCR and Western blotting were performed to detect the expression levels of *SLC7A7* and treating PSCs with DAAs reduced the expression levels of SLC7A7 mRNA and protein, which was consistent with previous results (Figure 3H–J). Overall, above results all suggest that stress is able to reduce the expression of the amino acid transporter *SLC7A7* and DEX can modulate the level of m6A modification of *SLC7A7* mRNA to affect its expression.

### 2.4. Effect of SLC7A7 on Protein Synthesis in Porcine Skeletal Muscle Cells

Previous results have shown that DEX affects the expression of protein synthesis-related genes in PSCs, so we further investigated the effect of *SLC7A7* on protein synthesis in PSCs by overexpression and interference of *SLC7A7*. The overexpression and interference of *SLC7A7* has been constructed successfully in our research work (Figure 4). After transfection of *SLC7A7* overexpression vector and interference fragment into PSCs, we found that the concentration of the protein was significantly increased after overexpression of *SLC7A7* compared with the control PCMV-HA group and the protein concentration was dramatically downregulated after interference of *SLC7A7* compared with the control NC group (Figure 5A,D). We also used a SUnSET approach and found that the rate of cellular protein synthesis upregulated upon overexpression of SLC7A7 to PSCs but dramatically downregulated after perturbing *SLC7A7* in cells (Figure 5B,C,E,F). The above results demonstrate that the amino acid transporter *SLC7A7* is able to promote protein synthesis in PSCs.

To further explore the role of *SLC7A7* in PSCs, we examined the expression levels of genes affecting protein synthesis using QPCR after transfection of a *SLC7A7* overexpression vector into PSCs, and the results showed that *MSTN*, *Atrogin-1*, and *FoxO1*, genes which reduce the protein synthesis, were decreased after overexpression of *SLC7A7*. However, the expression of *IGF1*, a gene promoting protein synthesis, was significantly upregulated (Figure 6A). Similarly, we transfected interfering fragments of *SLC7A7* into PSCs and found an elevated expression of *MSTN*, a gene that promotes proteolysis, but a decrease of *IGF1*, a gene that promotes protein synthesis (Figure 6B). mTOR signaling pathway is a key regulator in the process of protein synthesis, and it promotes total protein synthesis by phosphorylating two key downstream proteins S6K1 and 4EBP1. After overexpression of *SLC7A7*, it was found that the phosphorylation levels of mTOR and *S6K1* were significantly upregulated *SLC7A7*(Figure 6C,D) while significantly downregulated after transfected with *SLC7A7* interference fragment, and the phosphorylation level of 4EBP1 was also significantly downregulated (Figure 6E,F). These results suggest that amino acid transporter *SLC7A7* can increase cellular protein synthesis through the mTOR signaling pathway.

## 3. Discussion

Stress is an inevitable physiological response in the animal body in the face of detrimental environmental changes. Once the stress stimulus is detected during the development of the animals, a large cascade of GCs will be released through the HPA axis, which will promote the catabolism of protein and fat and affect the growth and development of skeletal muscle, making the production performance of livestock decline sharply [19,20,21]. Our research treated 0.1 µmol/L DEX to PSCs in 24 and 48 h to imitate the effect of GCs emerged in the animal body when they are confronting stress conditions, thus to find a universal indication about how GCs affects protein degradation and synthesis. Skeletal muscle serves as the largest protein repository and its quality is dependent on protein and cell turnover [22]. In the event of stress, GCs decrease the level of muscle protein synthesis and increase the level of protein degradation, ultimately leading to muscle atrophy [23]. GCs affect muscle protein synthesis through a variety of genes and pathways, among which GCs mainly through the UPS system, resulting in proteolysis [24]. DEX treatment of C2C12 cells, L6 myotubes or direct application of DEX to mice have all been reported to increase the expression of genes associated with muscle atrophy like *Atrogin-1* and *MuRF-1*, and activate the ubiquitin proteasome pathway, which promotes the breakdown of skeletal muscle proteins [25,26]. Previous research has found that puromycin binding efficiency was significantly reduced after treatment of C2C12 cells with DEX by using a non-set non-radioactive technique that detects protein synthesis rates in skeletal muscle cells, showing the anti-synthesis effect of DEX on muscle proteins [27]. In our work, it demonstrated similarly that stress evoked highly significant upregulation of atrophy related genes *Atrogin-1* and *MuRF1* in PSCs as well as significant decreases in the protein synthesis efficiency (Figure 1A–D), which indicated that stress brought about increased degradation and decreased synthesis rates of skeletal muscle proteins. mTOR is critical for maintaining muscle mass and function during skeletal muscle growth and preventing muscle wasting in mice [28,29]. Excessive glucocorticoid treatment is responsible for the reduction of protein synthesis levels in PSCs by inhibiting *mTOR* to *S6K1* signaling pathway [30,31,32]. GC-induced leucine zipper (GILZ), a GC-target protein to mediate several actions of GCs, has been found to promote apoptosis by inhibition of the Akt/mTOR signaling pathway in myeloma cells [33]. Since the *mTOR* signaling pathway is mainly studied in our research and essential in protein synthesis process, GILZ may also bind to mTORC2 to inhibit phosphorylation of AKT (at Ser473) and activate FoxO3a-mediated transcription of the pro-apoptotic protein BIMIN to inhabit protein synthesis procedure under stress condition. In L6 myoblasts, GCs inhibited IGF1 secretion and the phosphorylation levels of 4EBP1 and P70S6K1, which play key roles in protein synthesis, thereby reducing skeletal muscle protein synthesis [34]. In our study, the phosphorylation levels of mTOR, P70S6K, and 4EBP1 were all significantly decreased after DEX treatment in PSCs, indicating that GCs reduce the protein synthesis levels in skeletal muscle by downregulating mTOR signaling pathway. (Figure 1E,F).

Some epigenetic mechanisms, such as DNA methylation and chromatin modifications, are frequently involved in the regulation of gene expression in response to stimuli in an organism or under pathological conditions [35]. Among them, m^6^A modification is a part of the RNA epigenetics field, with evidence from studies that changes were found in m^6^A modification levels in vivo upon organismal stress [13]. As the body develops heat stress, the m^6^A modification of RNA is significantly higher in sheep than a control group, and *ALKBH5* promotes RNA demethylation in mammary cells when subjected to hypoxic stress [36,37]. Treatment of PSCs with DEX in our assay resulted in a decrease in overall intracellular m^6^A levels (Figure 2A,B). As a dynamic and reversible internal modification, m^6^A mainly plays a dynamic regulatory role through three enzymes: methylase, demethylase and reader protein. Previous research work found that METTL3 and METTL14 affect the process of cell differentiation by regulating the expression level of *MyoD* during mouse myoblast differentiation process [38,39]. A role of GR-mediated transcriptional regulation of m^6^A metabolic genes on m^6^A-depenent post-transcriptional activation of lipogenic genes in the chicken has been described in the literature [40], indicating that m^6^A/methylation genes were involved in GR pathways. The above studies both pointed out that m^6^A/methylation genes were affected by GR and the expression of subsequent GR-related genes in stress events [41]. As for our results, the mRNA and protein expression levels of METT3 and METTL14 were significantly downregulated in PSCs after DEX treatment (Figure 2C–E). It is tentatively concluded here that stress may affect the process of cell growth and development by affecting the expression levels of METTL3 and METTL14 in PSCs. However, the specific mechanism of how DEX affects the level of m^6^A modification in the cells needs further study.

GCs suppress protein synthesis to induced muscle atrophy through a variety of mechanisms, among which the catabolism of GCs inhibits the transport of amino acids into muscle cells, thereby reducing protein synthesis [34,42]. Cells will have an increasing demand for amino acids to satisfy the body’s needs in special events like cancer or stress which are detrimental to cell growth or development [43,44]. It has been reported that heat stress and exercise-induced stress responses can alter amino acid digestibility and the expression of amino acid transporters, export amino acids from muscle cells by increasing the expression of amino acid transporters [44,45,46]. In our results, the expression of *SLC7A7*, an amino acid transporter, was significantly downregulated in samples from cortisol-fed piglets and in PSCs treated with DEX (Figure 3A–D). The results above suggest that stress can affect the expression of the corresponding amino acid transporters in skeletal muscle, leading to reduced protein synthesis and loss of muscle mass. DEX causes significant changes in m^6^A modification levels in PSCs (Figure 2), so are there any such modifications at the *SLC7A7* gene as well? The m^6^A post transcriptional modification mainly occurs in response to stress by regulating gene expression through a series of RNA metabolic ways that affect genes mRNA translation and degradation [47]. It has been shown that amino acid transporters can be regulated by m^6^A modification, as the main glutamine transporter *SLC1A5* in cancer cells can be regulated by the m^6^A modified demethylase *FTO* and *SLC7A11* is promoted to decay by preferentially bound to *YTHDC2* [48]. We detected a highly significant downregulation of *SLC7A7* mRNA m^6^A modification compared with the control (Figure 3E,F), and the expression level of *SLC7A7* was significantly downregulated after treatment with DAA, an inhibitor of m^6^A methylation. These results confirm that *SLC7A7* is subject to m^6^A posttranscriptional modification after glucocorticoid treatment (Figure 3G–J). Also, the m^6^A modified reader protease YTHDF2 was significantly upregulated in cells with DEX treatment (Figure 2). Thus, we hypothesized that the degradation of *SLC7A7* might be controlled by YTHDF2 in an m^6^A-dependent manner.

Amino acid transporters can act the upstream of the mTOR signaling pathway, enabling cells to sense the availability of amino acids and initiate anabolic responses [49]. Studies have shown that leucine is required for activation of the mTOR signaling pathway, which is mainly transported by members of SLC7 family of cationic amino acid transporter solute carriers, among which *SLC7A7* can transport leucine intracellularly, and it has also been shown that *SLC7A7* can regulate the mTOR signaling pathway and affect the phosphorylation levels of the downstream effector S6K1 [50]. In our study, after overexpression of *SLC7A7*, the synthesis levels of cellular proteins were found to be upregulated, and the phosphorylation levels of mTOR signaling pathway and its two downstream S6K1 and 4EBP1 were also significantly upregulated; in addition, the opposite results were obtained after the interference with *SLC7A7* in cells (Figure 4, Figure 5 and Figure 6), implying that *SLC7A7* is required for PSCs protein synthesis by regulating the signaling of mTOR pathway.

In summary, when animals fall into a stress situation, the body’s HPA axis will be activated, and the level of GCs increases, making mTOR signaling significantly downregulate. Furthermore, when stress occurs, the overall m^6^A modification level of the skeletal muscle cell would significantly downregulate, resulting in a significant decrease of mRNA and protein expression of *SLC7A7* and mTOR signaling pathway, and finally decrease total protein synthesis in porcine skeletal muscle (Figure 7).

## 4. Materials and Methods

### 4.1. Animals and Samples

The Yorkshire piglets of 1–2 days were provided by the Farm of Huazhong Agricultural University (Wuhan, China). After the piglets were slaughtered, the longissimus dorsi and leg muscles were taken to separate for the primary skeletal muscle cells. Finally, the skeletal muscle cells were grown in incubators at 37 °C and 5% CO_2_, and proliferating cells were cultured in Dulbecco’s Modified Eagle’s Medium (DMEM) supplemented with 10% fetal bovine serum (DMEM, Gibco, Grand Island, NY, USA). The 28-day-old Yorkshire pigs were fed with cortisol in 120 mg/kg diet for 7 days to mimic stress, then the *longissimus dorsi* (LD) muscle was sampled and analyzed for gene expression, and more detailed information can be found in our previous work [50].

Animals: all experimental animal procedures in this study were performed according to the guidelines of Good Laboratory Practice, and the animals were supplied with nutritional food and sufficient water. Animal feeding and tests were conducted based on the National Research Council Guide for the Care and Use of Laboratory Animals and approved by the Institutional Animal Care and Use Committee at Huazhong Agricultural University.

Isolation and culture of primary skeletal muscle cells: primary skeletal muscle cells were isolated from the longissimus dorsi skeletal muscles of 2–3 day old piglets, minced and digested in a mixture of type I collagenase and DMEM (DMEM, Gibco, Grand Island, NY, USA).

### 4.2. Total RNA Preparation and cDNA Synthesis

Total RNA was isolated at 48 h after cell transfection, using total RNA extraction kit (Omega bio-tek, Norcross, GA, USA) according to the manufacturer’s protocol. The RNA integrity was checked using denaturing gel electrophoresis and the RNA concentration was measured with a NanoDrop 2000 spectrophotometer (Thermo Scientific, Waltham, MA, USA). Total RNA was reverse transcribed using a RevertAidTM First Strand cDNA Synthesis Kit (Thermo Scientific, Waltham, MA, USA).

### 4.3. Quantitative Real-Time Polymerase Chain Reaction (PCR)

The specific fluorescent quantitative PCR primers were designed using cDNA as a template; the mRNA level was quantified using β-actin gene as an internal reference; the real-time quantitative PCR experiments were performed on the CFX384 TouchTM fluorescence quantitative PCR instrument using SYBR Green qPCR Master Mix (Bio-Rad). Finally, the 2^−ΔΔ^Ct method was used for data analysis, and one-way analysis of variance was performed to determine the significance at *p* < 0.05 (significant) and *p* < 0.01 (extremely significant). Primer sets are listed in Table 1.

### 4.4. BCA Method for Detecting Protein Concentration

We used the BCA Protein Concentration Test Kit (Thermo Fisher America) to perform detection according to the procedure.

### 4.5. Plasmid Construction, siRNA Synthesis and Cell Transfection

Briefly, For the *SLC7A7* overexpression plasmids, full-length sequences were cloned into the pCMV-HA plasmid. Full-length GRα and SLC2A4 sequences were amplified with full-length-F/R primers (Table 2). LC7A7 siRNA was synthesized by Shanghai GenePharm Company, using the following *SLC7A7* siRNA sequences: *SLC7A7* siRNA (sense): CCUACAUCCUCGAGGCCUUTT; *SLC7A7* siRNA (anti-sense):

AAGGCCUCGAGGAUGUAGGTT. For cell transfection, primary cells were transfected with 4 μg of expression vectors or 3.3 μg/μL of siRNA oligo using Lipofectamine 3000 (Invitrogen, Carlsbad, CA, USA) in each well of a 6-well plate.

### 4.6. Western Blot Analysis

Briefly, PSC was washed with PBS and lysed in RIPA lysis buffer (Beyotime Bio-technology Company, Shanghai, China). Next, 20 μg of total protein was resolved by 10% sodium dodecyl sulfate-polyacrylamide gel electrophoresis and electro-transferred onto a poly-vinylidene fluoride (PVDF) (Millipore, Burlington, MA, USA) membrane. The PVDF membrane was blocked in 5% skim milk powder dissolved in TBST for 90 min at room temperature. Primary antibodies were applied in sealing fluid at 4 °C overnight. Subsequently, the PVDF membrane was washed with TBST and stained with the appropriate HPR-labeled secondary antibodies (goat anti-rabbit or mouse) for 1 h at room temperature. After washing with TBST, the membrane was analyzed using the ECL Reagent (Beyotime Biotechnology). The antibodies used included: GAPDH (GB11002) antibody purchased from Servicebio, Wuhan, China; β-Actin (AC037), MSTN (A6913), FOXO1 (A7637), Atrogin-1 (MAFbx) (A3193) and secondary antibody purchased from Abclonal Technology, Wuhan, China, *SLC7A7* (NBP1-59856, Novusbio); m^6^A(202003,SynapticSystems); p70S6K1 (sc-230, Santa); P-S6K1 (AF3228, Affinity Biosciences), mTOR (2971S, Cell Signaling Technology); P-mTOR(S2448, Cell Signaling Technology); 4EBP1 (RT1002,HUABIO); p-4EBP1 (RT1004, HUABIO) and mouse puromycin antibody 12D10 (320591, Millipore).

### 4.7. Surface Sensing of Translation (SUnSET) Non-Radioactive Method for Detecting Protein Synthesis Rate

In this study, the surface sensing of translation (SUnSET) non-radioactive method was used to measure the protein synthesis rate of porcine skeletal muscle cells. After transfection with the *SLC7A7* overexpression vector and *SLC7A7* interference fragment, the cells were treated separately with DEX and RU486 for 48 h, followed by the addition of 1 μg/mL puromycin (Beyotime Biotechnology, Shanghai, China) and treatment of 30 min. Finally, the change of puromycin was detected using a Western blot.

### 4.8. m6A Immunoprecipitation (MeRIP) qPCR

1 μg m6A antibody (Abcam, ab151230) with 20 μL beads A/G (Bio-rad, 161-4013; 161-4023) Incubate overnight at 4 °C. This was followed by 1 × IP buffer containing 40 U RNase inhibitors (50 mM Tris · HCl, 750 mM NaCl, and 0.5% NP40) Wash the above beads, add 100 μg of fragmented mRNA, respectively, and continue the incubation overnight. Then, we discarded the solution and added the eluent containing 8.4 μg proteinase K (5 mM Tris · HCl, 1 mM EDTA, and 0.05% SDS) then incubated it at 50 °C for 1.5 h. Finally, phenol chloroform was used to extract RNA and combine qPCR to detect the expression of related genes.

### 4.9. Statistical Analysis

The results are presented as means ± standard deviation (SD). Statistical analysis of the groups was performed using Student’s *t*-test or one-way ANOVA with the LSD post hoc test. Statistical significance was set at * *p* < 0.05 and ** *p* < 0.01.

## Figures and Tables

**Figure 1 ijms-23-00661-f001:**
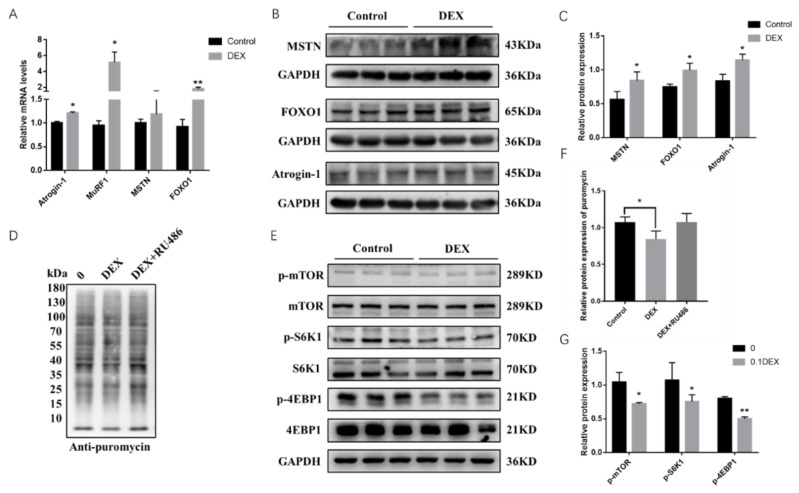
Glucocorticoids (GCs) increase protein breakdown and decrease protein synthesis in porcine skeletal muscle cells. (**A**) The mRNA expression levels of *Atrogin-1*, *MuRF1*, *FOXO1* and *MSTN* treated with DEX; (**B**) the protein expression levels of Atrogin*-1*, FOXO1 and MSTN treated with DEX; (**C**) quantification of Western blotting under the treatment of DEX; (**D**) effect of glucocorticoid and its antagonist on the protein synthesis rate; (**E**) the phosphorylation level of mTOR, S6K1 and 4EBP1 in mTOR signaling pathway after adding DEX; (**F**) quantification of protein synthesis rate under the treatment of DEX and RU486; (**G**) quantification of Western blotting under the treatment of DEX. *, ** indicate significant difference at *p* < 0.05 and *p* < 0.01, respectively.

**Figure 2 ijms-23-00661-f002:**
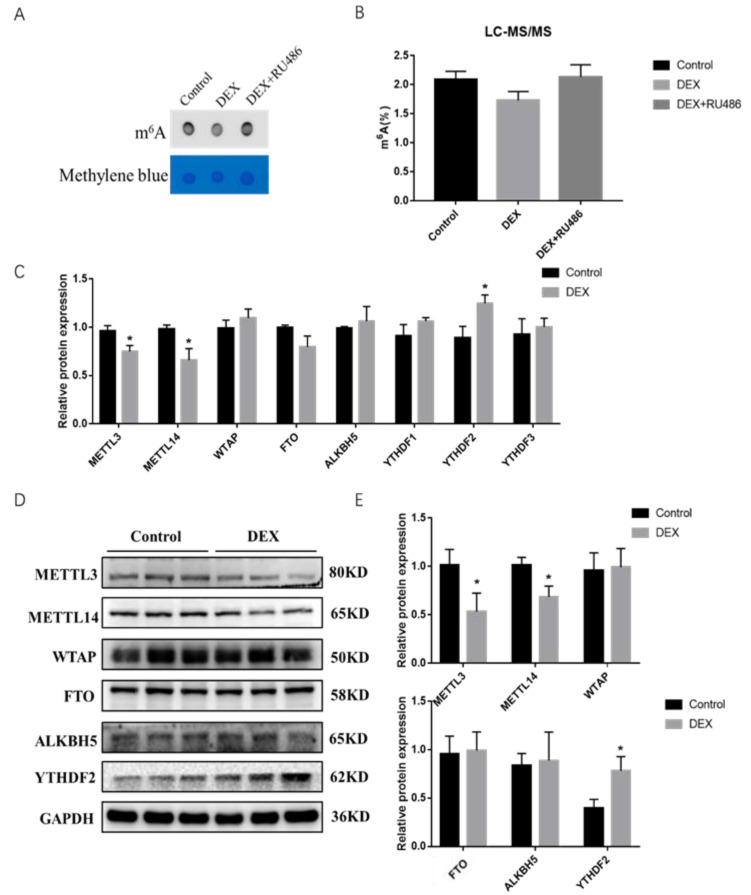
m^6^A modification level changes after DEX and RU486 treatment in porcine skeletal muscle cells. (**A**) Dot blot confirmed the m^6^A modification after DEX and RU486 treatment; (**B**) the m^6^A ratio in RNA was measured using LC-MS/MS; (**C**) the mRNA expression levels of *METL3*, *METL14*, *WTAP*, *FTO*, *ALKBH5* and *YTHDF1-3* after treatment with DEX; (**D**) protein expression levels of METTL3, METTL14, WTAP, FTO and YTHDF2 after treatment with DEX; (**E**) quantification of Western blotting under the treatment of DEX. * Indicates a significant difference at *p* < 0.05.

**Figure 3 ijms-23-00661-f003:**
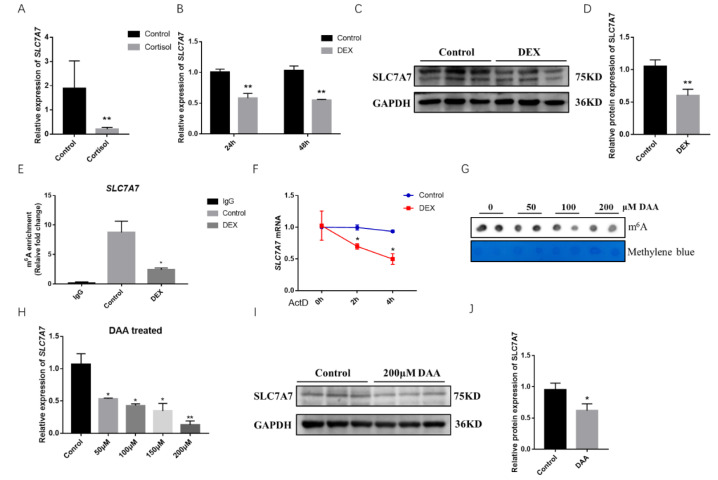
Effect of GCs on the amino acid transporter *SLC7A7*. (**A**) The mRNA expression levels of *SLC7A7* in longissimus dorsi muscle of the cortisol group and the control group, respectively; (**B**) the mRNA expression levels of *SLC7A7* after DEX treatment for 24 h and 48 h; (**C**) the protein expression levels of SLC7A7 after DEX treatment; (**D**): quantification of Western blotting under the treatment of DEX; (**E**) validation of m^6^A modification in *SLC7A7* mRNA by MeRIP-QPCR; (**F**) the mRNA expression of *SLC7A7* was detected at 0, 2 and 4 h after ActD treatment; (**G**) Dot blot detection of m^6^A modification levels after treatment with 0, 50, 100 and 200 μM DAA; (**H**) the mRNA expression level of *SLC7A7* after treatment with 0, 50, 100 and 200 μM DAA; (**I**) the protein expression levels of SLC7A7 after treatment with 200 μM DAA; (**J**) quantification of Western blotting under the treatment of DAA. *, ** indicate significant difference at *p* < 0.05 and *p* < 0.01, respectively.

**Figure 4 ijms-23-00661-f004:**
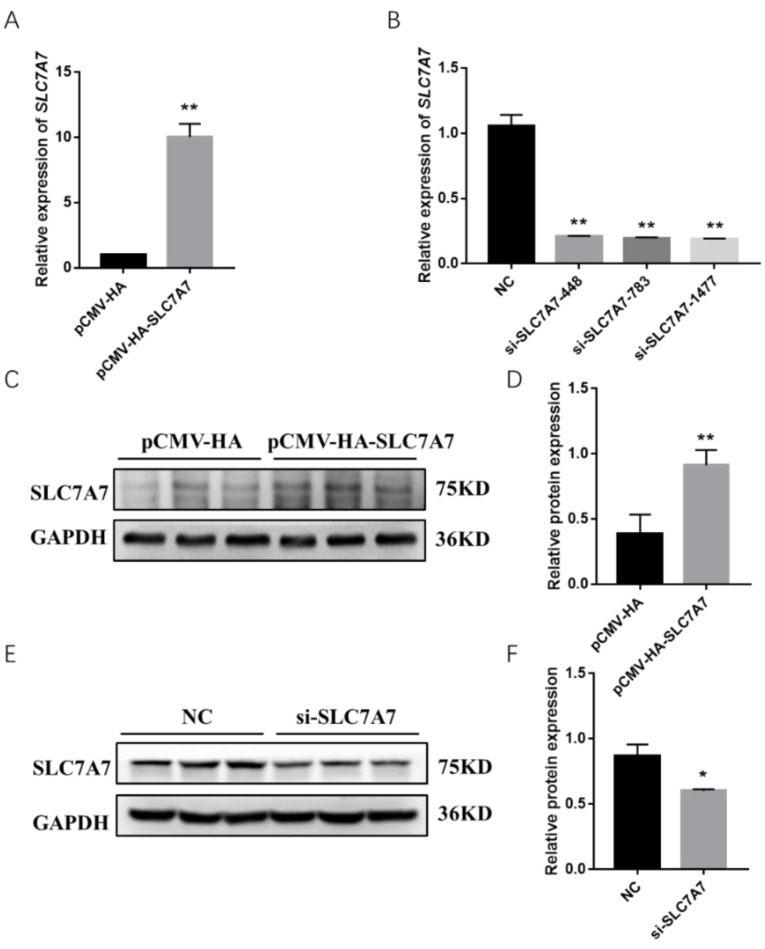
Expression level changes following overexpression and interference of *SLC7A7* in PSCs. (**A**) The mRNA expression level of *SLC7A7* after overexpression of *SLC7A7*; (**B**) the mRNA expression level of *SLC7A7* after interference of *SLC7A7*; (**C**) protein expression level of SLC7A7 after overexpression of *SLC7A7*; (**D**) Quantification of Western blotting after overexpression of *SLC7A7*; (**E**) the protein expression level of SLC7A7 after interference of *SLC7A7*; (**F**) quantification of Western blotting after interference of *SLC7A7*. *, ** indicate significant difference at *p* < 0.05 and *p* < 0.01, respectively.

**Figure 5 ijms-23-00661-f005:**
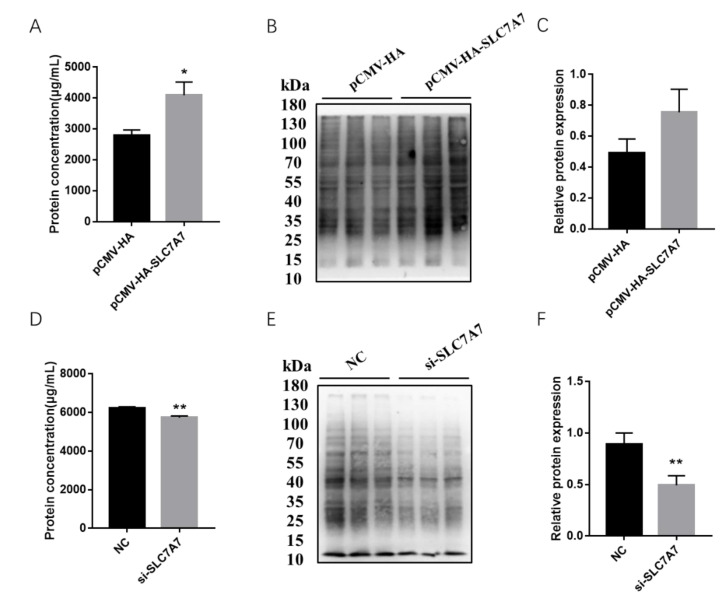
Overexpression and interference of *SLC7A7* affect cell protein synthesis. (**A**) Total cell protein after overexpression of *SLC7A7*; (**B**) the effect of overexpression of *SLC7A7* on the rate of protein synthesis; (**C**) protein quantification of puromycin; (**D**) total cell protein after interfering of *SLC7A7*; (**E**) the effect of interfering of *SLC7A7* on the rate of protein synthesis; (**F**) protein quantification of puromycin. *, ** indicate significant difference at *p* < 0.05 and *p* < 0.01, respectively.

**Figure 6 ijms-23-00661-f006:**
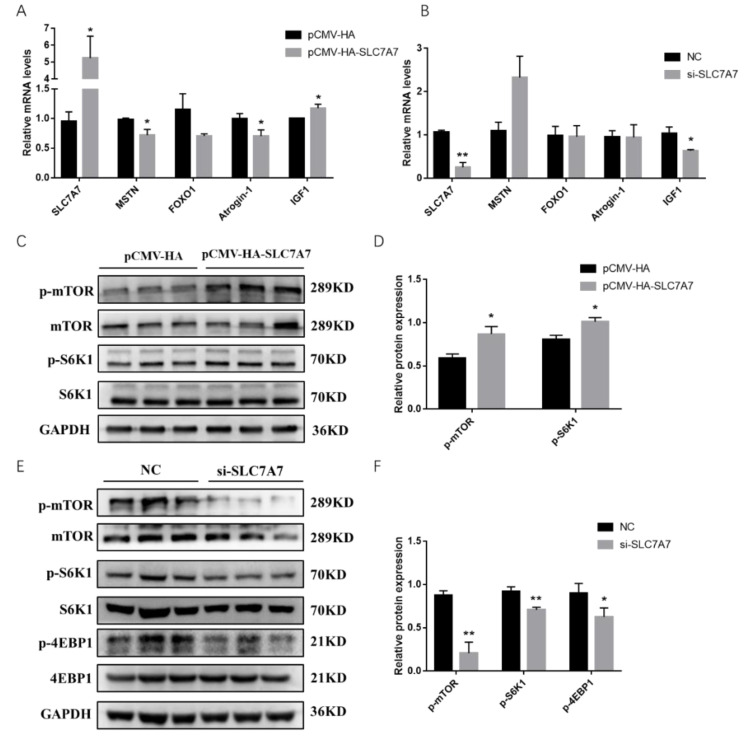
Overexpression and interference of *SLC7A7* in PSCs affect expression of genes involved in protein synthesis and mTOR signaling pathway. (**A**) The mRNA expression level of *SLC7A7*, *MSTN*, *FOXO1*, *Atrogin-1* and *IFG1* after overexpression of *SLC7A7*; (**B**) the mRNA expression level of *SLC7A7*, *MSTN*, *FOXO1*, *Atrogin-1* and *IFG1* after interference with *SLC7A7*; (**C**) the phosphorylation level of mTOR and S6K1 after overexpression of *SLC7A7*; (**D**) quantification of Western blotting after overexpression of *SLC7A7*; (**E**) the phosphorylation level of mTOR, S6K1 and 4EBP1 after interference with *SLC7A7*; (**F**) quantification of Western blotting after interference with *SLC7A7*. *, ** indicate significant difference at *p* < 0.05 and *p* < 0.01, respectively.

**Figure 7 ijms-23-00661-f007:**
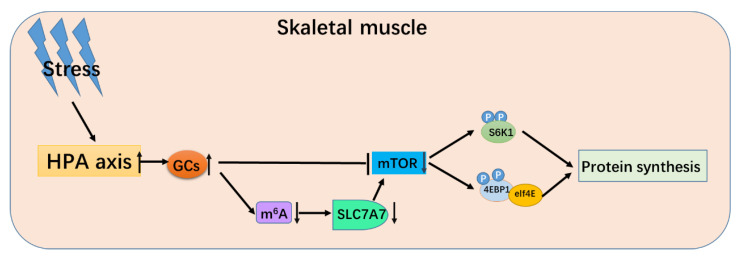
The schematic diagram of glucocorticoid regulating protein deposition process.

**Table 1 ijms-23-00661-t001:** Primer sequences used for quantitative real-time polymerase chain reaction (qPCR).

Gene	Sequence of Primer (5′–3′)	Size (bp)
*METTL3*	F:CTGGGGTTACGAACGGGTAG	243
	R:CCAGGGGACAGTCTCTCAATC	
*METTL14*	F:TGTGTTTACGCAAGTGGGGT	179
	R:AATGAAGTCCCCGTCTGTGC	
*WTAP*	F:GGCCAACGGACCAAGTAATG	246
	R:TCATGTGAGTGGCGTGTGA	
*FTO*	F:CCCCAGAAAATGCCGTACCT	228
	R:ACCAGGGGTCTCTATGTCCC	
*ALKBH5*	F:CGTGTCCGTGTCCTTCTTCA	201
	R:AGGATGATGACAGCTCTGCG	
*YTHDF1*	F:CAAGTGGAAGGGCAAGTTCG	182
	R:AGTGCTTGTAGGAGGCGATG	
*YTHDF2*	F:ATGCCTCGGCCATTGTGTG	211
	R:CGCCGAGAGAAGGGAACAC	
*YTHDF3*	F:GCGCTTCGCCTTCAAGTGTA	220
	R:TTCAGGGAACGGTAAGCTGC	
*SLC7A7*	F:TGTGCCTATGTCAAGTGGGG	171
	R:CAGAGCGATGTCACCTGTTG	
*IGF1*	F:CTCTCCTTCACCAGCTCTGC	200
	R:TCCAGCCTCCTCAGATCACA	
*MuRF1*	F:ATGGAGAACCTGGAGAAGCA	219
	R:ACGGTCCATGATCACCTCAT	
*β-actin*	F:CCAGGTCATCACCATCGG R:CCGTGTTGGCGTAGAGGT	155

**Table 2 ijms-23-00661-t002:** *SLC7A7* CDS region amplification primer sequence.

Gene	Sequence of Primer (5′-3′)	Size (bp)
*SLC7A7*	F:**CCGGAATTC**ATGGTTGACGGCATGAAGTA	1542
	R:**CTAGCTAGC**TTAATTAGACTTGGGATCTT	

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
