# Peer review of "Glucocorticoid Regulates the Synthesis of Porcine Muscle Protein through m6A Modified Amino Acid Transporter SLC7A7"

_ijms, 2022, doi:10.3390/ijms23020661_

Round 1

Reviewer 1 Report

The data included in this manuscript are compelling overall. However I have several concerns that need to be addressed regarding the details and content, which I have outlined below. 

Major concerns:

  • Please quantify puromycin protein synthesis (SUnSET assay) and include in figures.
  • Methodological details are missing or need clarification
    • Species of pig used
    • There is nothing in the methods regarding the cortisol-fed pig experiment shown in Figure 3A. Please include concentration and duration of cortisol, age of pigs, etc.. 
    • MeRIP-qPCR not included in methods--need details of experiment
    • Need to include final concentration of siRNA used, not volume 
  • Substantial grammatical edits are needed throughout.
    • In addition, please attend to the following:
      • Please reduce repetitive statements throughout.
      • Intro rationale and flow needs work- please improve clarification for the reader as to why you are looking into m6A/methylation genes.
  • Please discuss how closely your treatment paradigm compares to that of endogenous circulating cortisol due to stress in pigs.
  • There are a few statements throughout that do not have any citations, please include. 

Minor concerns:

  • Nomenclature of genes and proteins should be consistent throughout
  • Data interpretation is included in the results section 
  • In line 228, you sate that 'our study found that puromycin binding efficiency...treatment of DEX on C2C12 cells' I did not see anything about using C2C12s in the methods or figures. Please clarify if this was a previous study, if so please cite that study (the citations included are not from your group), if not please provide the details of this experiment or correct any mistakes.

Reviewer 2 Report

GLUCOCORTICOID REGULATES THE SYNTHESIS OF PORCINE MUSCLE PROTEIN THROUGH M6A MODIFIED AMINO ACID RANSPORTER SLC7A7

We-Jing Xu and colleagues studied the effect of dexamethasone on the protein synthesis level of porcine skeletal muscle cells.

Since glucocorticoid-induced leucize zipper (gilz) is an early glucococrticoid induced gene, i would ask the authors to see if gilz i salso involved in the patways they studied.

Round 2

Reviewer 1 Report

I appreciate the authors addressing all of my concerns in detail and revising the manuscript as suggested.
